# The Effect of Employee-Oriented Flexible Work on Mental Health: A Systematic Review

**DOI:** 10.3390/healthcare10050883

**Published:** 2022-05-10

**Authors:** Rahman Shiri, Jarno Turunen, Johanna Kausto, Päivi Leino-Arjas, Pekka Varje, Ari Väänänen, Jenni Ervasti

**Affiliations:** Finnish Institute of Occupational Health, P.O. Box 18, Työterveyslaitos, FI-00032 Helsinki, Finland; jarno.turunen@ttl.fi (J.T.); johanna.kausto@ttl.fi (J.K.); paivi.leino-arjas@ttl.fi (P.L.-A.); pekka.varje@ttl.fi (P.V.); ari.vaananen@ttl.fi (A.V.); jenni.ervasti@ttl.fi (J.E.)

**Keywords:** anxiety, depression, mental disorders, telecommuting, schedule control, worktime control, work schedule

## Abstract

The effect of flexible work on mental health is not well known. The aim of this systematic review was to assess the effects of employee-oriented flexible work on mental health problems and associated disability. Literature searches were conducted in the PubMed, Scopus, Web of Sciences, Cochrane Library, PsycINFO, ProQuest and EconPapers databases from their inception through October-November 2020. Sixteen studies on the associations of worktime control, working from home, or flexible working arrangements with mental health related outcomes were included in the review: one cluster randomized controlled trial, two non-randomized controlled trials, two cross-over studies, and 11 prospective cohort studies. Three reviewers independently assessed the met-hodological quality of the included studies and extracted the data. The included studies differed in design, intervention/exposure, and outcome, so meta-analysis was not carried out and qualitative results were reported. A few prospective cohort studies found that low employees’ control over worktime increases the risk of depressive symptoms, psychological distress, burnout, and accumulated fatigue. One cross-over and a few cohort studies found small beneficial effects of working partly from home on depressive symptoms, stress, and emotional exhaustion. A small number of controlled trials, cross-over or cohort studies found that flexible working arrangements increase employees’ control over working hours, but have only modest beneficial effects on psychological distress, burnout, and emotional exhaustion. This systematic review suggests that employee-oriented flexible work may have small beneficial effects on mental health. However, randomized controlled trials and quasi-experimental studies are needed to identify the health effects of flexible work.

## 1. Introduction

Globally, the COVID-19 pandemic markedly increased working from home, at least temporarily [1]. Working from home as an example of telework is part of flexible working arrangements. Some previous studies have reported that flexible working arrangements are linked to better balance of work and private life, and consequently lead to longer work careers [2,3]. However, other studies have reported that flexible working arrangements may increase work–private life conflict [4], and have no beneficial effects on health [5].

Depending on the organization, flexibility may entail flexibility in time, in space, or in the way the work is performed [6]. Flexible work comprises company-oriented and employee-oriented flexibility. Company-oriented flexibility involves employers extending, modifying, or reducing working hours or other work arrangements according to organizational objectives, for example, to better meet a financial budget. Employee-oriented flexible work permits workers to modify when, where, or how to work. It can refer to autonomy regarding working times (i.e., worktime control) or working location or other work arrangements [7]. Worktime control refers to control over work schedule, or control over working hours and non-working hours (e.g., when to have holidays or unpaid leave) [8]. Employer-based flexibility has been linked to employees’ poor health and wellbeing, while employee-oriented flexibility has been associated with better employee health and wellbeing [9,10].

Mental disorders have surpassed musculoskeletal disorders as the leading cause of disability benefits in many EU countries [11,12], and are among the leading causes of years lived with disability [13,14]. The number of years lived with disability due to depressive disorders increased by 32% in females and by 35% in males between 1990 and 2007 and further increased by 14% in females and by 15% in males between 2007 and 2017 [14]. Depressive disorders were the third leading cause of years lived with disability in females and the fifth leading cause in males in 2017 [14]. However, mental health also entails the positive side: improving mental wellbeing increases work engagement [15]. Some studies suggested that flexible working arrangements decrease symptoms of mental disorders and improve mental health [16,17,18]. Enabling employees to determine their own working patterns more freely can boost their mental well-being and protect against mental health decline. Given the widespread use of flexible work arrangements in various sectors of employment, the potential role of flexible working arrangements in the prevention and management of mental health problems is noteworthy due to the steep rise in mental health problems in various countries. In the presence of a positive association between flexible working arrangements and mental health, even a small beneficial effect can produce considerable impacts in current societies where there are more options to arrange the time and place of work.

To date, three systematic reviews have been published on the association between employee control over working hours or flexible working hours and health. A systematic review [19] published in 2007 examined the associations of organizational-level measures to improve employee control or participation in workplace decision-making with psychosocial factors and health. The review included 18 studies published between 1981 and 2006 on various workplace measures such as control over working hours, flexible working hours, identifying and reducing workplace stressors, establishing problem-solving committees, a health-related educational course, stress management training, and participatory ergonomic changes. The review found that a decrease in demands or increase in employee control or support had beneficial effects on health, particularly mental health. Another systematic review published in 2012 included studies on the health effects of worktime control published between 1995 and 2011 [20]. Of the 53 studies included in that review, the majority were cross-sectional studies, and only six studies were prospective cohort studies and five were experimental studies. The review found that worktime control improved work-private life balance and found evidence only for a cross-sectional association between control over daily working hours and health/wellbeing. Furthermore, a Cochrane systematic review on the health effects of flexible working conditions [21] published in 2010 included four non-randomized controlled trials on self-scheduling of shift work and one trial on flexible working time. The review [21] found that self-scheduling of shift work had beneficial health effects, while a single before-after study on flexible starting and ending times of a workday and flexible timing and duration of lunch breaks found no health effects. In the earlier reviews, the association between employee-oriented flexible work and mental health was mainly based on cross-sectional studies. Furthermore, the previous reviews did not examine the association between working from home and mental health.

To date, the role of flexible work in the prevention of mental health problems or to support mental health is not well known. In the present study, we aimed to systematically review the available literature on the effects of employee-oriented worktime control, working from home, flexible work (when, where, or how to do the work), and employee-oriented flexible workplace arrangements on mental health, mental disorders, and work disability due to mental disorders.

## 2. Methods

### 2.1. Search Strategy

We used the PRISMA statement [22] to develop the review protocol. The checklist consists of 27 recommendations which help the reviewers to report transparently the results of their systematic reviews. The review protocol was registered in PROSPERO (CRD42020214134). We searched PubMed, Scopus, Web of Sciences, Cochrane Library, PsycINFO, ProQuest and EconPapers databases from their inception through October-November 2020. Appendix A shows the search terms and search strings used in the different databases. The language of publications was restricted to English only. The reference lists of included articles were also hand-searched for any additional relevant studies. Our search strings lacked a search term “worktime control”. We conducted an additional search for “worktime” OR “work-time” in relation to mental health problems in PubMed and Embase.

### 2.2. Inclusion and Exclusion Criteria

For each publication, at least two reviewers (J.T., J.E., R.S., J.K., P.L.-A., P.V., or A.V.) independently screened the titles and abstracts using Covidence to identify studies on the associations of employee-oriented worktime control, working from home, flexible work (when, where, or how to do the work), and employee-oriented flexible workplace arrangements with any mental health problem and associated work disability (sickness absence or disability pension due to mental disorder) or indicators of positive mental health outcomes. Randomized and non-randomized clinical trials, cross-over studies, prospective cohort studies, and case-control studies were eligible. Cross-sectional studies were excluded due to their descriptive nature with no indication on cause and effect. Studies on the number of weekly working hours, studies on different work schedules or shift work, studies comparing long with short shifts, studies comparing part-time with full-time sickness absence, and studies on overtime work or extended work availability were not classified as employee-oriented flexible work and were excluded from the review. Disagreements between the reviewers were resolved by a third reviewer.

### 2.3. Quality Assessment

For each publication, three reviewers (J.T., J.E. and R.S.; or J.K., P.L.-A. and R.S.) independently evaluated the methodological quality of the studies included in the review using the Cochrane risk of bias tool (ROB-2) for randomized controlled trials [23], ROBINS-I for quasi-experimental studies [24], Joanna Briggs Institute’s Critical Appraisal Checklist for Cohort Studies [25], and used a checklist proposed by Ding et al. [26] based on the Cochrane handbook for cross-over studies. Disagreements between raters were resolved through discussion with or the involvement of an additional reviewer. The Cochrane risk of bias tool (ROB-2) evaluates bias arising from the randomization process, bias arising from the timing of randomization, bias due to deviations from intended interventions, bias due to missing outcome data, bias in measurement of the outcome, and bias in selection of the reported result.

### 2.4. Data Synthesis

We extracted the details of the studies included in the review such as year of publication, country, follow-up time, study population, sex distribution, age range and/or mean age, sample size, intervention or exposure, outcome, findings, and adjustment for confounding factors. The included studies differed in study design, intervention/exposure (worktime control, working from home/teleworking, or a combination of flexible working arrangements) and outcome (different indicators of symptoms of mental ill-health or positive mental health outcomes) and we therefore synthesized the results qualitatively.

## 3. Results

Our searches identified 1915 relevant publications in PubMed, 3767 in Web of Science, 3997 in Scopus, 1436 in Cochrane Library, 3078 in PsycINFO, 333 in ProQuestand and 421 in EconPapers (Table 1 and Figure 1). After removing 3961 duplicates, 8832 publications remained relevant to be screened. A total of 8748 reports were ineligible based on screening titles and abstracts for study design, population, exposures/interventions, and outcomes. The overall percentage of interrater agreement between the reviewers was more than 97%. Moreover, our additional search of “worktime” OR “work-time” in relation to mental health problems retrieved 231 publications in PubMed and 330 in Embase. Ninety-six studies were selected for full-text screening, and of them 16 studies (19 articles, the results of some studies have been reported in more than one article) including one cluster randomized controlled trial, two non-randomized controlled trials, two cross-over studies, and 11 prospective cohort studies (14 reports) fulfilled the inclusion criteria and were included in the review (Table 1 and Appendix A). Mental health outcomes included depressive symptoms, perceived stress, psychological distress, emotional exhaustion, burnout, accumulated fatigue, disability pension due to mental disorders, sickness absence due to depression or anxiety, and work-related well-being (work engagement). The risk of bias was high for the randomized controlled trial due to measurement of outcome and missing outcome data (Appendix A), moderate for the non-randomized controlled trials due to not adequately controlling for confounding factors, selection of participants and missing data (Appendix A), and moderate for cross-over studies (Appendix A). For prospective cohort studies, the number of “yes” scores (low risk of bias) ranged between 6 and 10 out of 11 items (Appendix A).

### 3.1. Worktime Control

A small (N = 35 for intervention group and n = 187 for control group) non-randomized controlled trial [28] found that worktime self-scheduling via a computer program increased employees’ involvement in the planning of their working hours but did not decrease the level of stress. Moreover, a cohort study found that schedule flexibility assessed with a single item was not associated with changes in depressive symptoms among working women who returned to work within six months after childbirth [39].

Most of the included prospective cohort studies examined the impact of worktime control on mental health outcomes. The included prospective cohort studies used between four and eight items to assess worktime control and the items consisted of perceived control over (1) length of a workday, (2) starting and ending times of a workday, (3) taking breaks during work, (4) handling private matters during work, (5) which days to work), (6) scheduling shifts, (7) scheduling vacations, and (8) scheduling unpaid leave [29,30,31,33,34,41].

High worktime control at both baseline and follow-up was associated with lower risk of depressive symptoms at follow-up [41], and low control over daily hours (i.e., length of a workday and starting and ending times) or low control over time off (i.e., scheduling vacation and unpaid leave) at baseline was associated with higher risk of depressive symptoms at follow-up [34]. The associations did not differ between men and women [34]. Moreover, this prospective cohort study [34] found a bidirectional association between low worktime control and depression. Low control over worktime was associated with subsequent depressive symptoms, and depressive symptoms were associated with subsequent low control over worktime. However, this study used prevalent cases rather than incident cases as outcomes of interest and did not exclude cases of depression at baseline when assessing the effect worktime on subsequent depressive symptoms and did not exclude people with low worktime control at baseline when assessing the effect of depressive symptoms on subsequent low control over worktime. Thus, a bidirectional association between low worktime control and depressive symptoms observed in the study [34] can be due to analysing a prospective cohort study like a cross-sectional study.

Work-life imbalance partly mediated the relationship between worktime control and depressive symptoms [35]. However, the indirect effect of low worktime control on depressive symptoms was small, and neither control over daily hours nor control over time off was associated with sickness absence due to depression or anxiety [29].

Two cohort studies found an association between low worktime control and increased risk of burnout [32,33]. One of these studies showed that the elimination of low worktime control can prevent 5–7% of burnout cases [32]. A cohort study (two reports) found an association between worktime control and mental health problems in women only. Low worktime control was associated with a higher risk of psychological distress [31], and high worktime control was associated with a lower risk of disability pension due to mental health problems among female dominated municipal employees (77.5 to 78.6% of the participants were women) [30]. However, the latter study [30] controlled the observed association for age and socioeconomic status only, and co-worker assessed worktime control was not associated with the risk of disability pension due to mental disorders.

Lastly, high worktime control at both baseline and follow-up was associated with lower risk of accumulated fatigue measured with 11 items and higher recovery from fatigue at follow-up [41]. Moreover, a small cohort study [38] subjectively assessed accumulated fatigue using 13 items, which included several items on symptoms of mental illness in the past month and objectively measured fatigue using a psychomotor vigilance task. The increase in worktime control during one-year follow-up was not associated with the subjective measure of accumulated fatigue, but it was associated with objectively measured fatigue [38].

### 3.2. Working from Home/Teleworking

A cohort study among 570 working women who returned to work within six months after childbirth found that women who worked from home had a lower risk of depressive symptoms than women who did not work from home; however, the number of hours worked from home was not associated with changes in depressive symptoms [39]. Another prospective cohort study found a U-shaped or J-shaped association between the number of hours worked from home per month and depression. Non-telecommuters were at a higher risk of depression than telecommuters [37]. Telecommuters who worked from home for eight hours or less per month had a lower risk of depression compared with non-telecommuters. The difference was not statistically significant for off-hour telecommuting and prime time telecommuting for 9–32, 33–72, and 73 h or more per month [37].

A small cross-over study (N = 39) [17] reported that the level of stress was reduced during one-week telework in a rural archipelago environment and did not return to the original level after the experiment. However, the beneficial effect for stress was small. Moreover, the study [17] found that emotional exhaustion decreased during telework but returned to the original level after the experiment, and telework had no effect on work engagement [17]. A cohort study found no statistically significant association between the number of hours worked from home per month and stress [37]. Another cohort study found that an increase in interpersonal interaction was associated with the increased risk of work exhaustion, and part-time telework (one or two days per week) reduced the effect of interpersonal interaction on work exhaustion [36].

### 3.3. Flexible Working Arrangements

The included studies defined flexible work arrangements in various ways. The intervention consisted of at least two domains, also including working from home and worktime control. A cluster randomized controlled trial [18] examined the effect of an intervention to facilitate working from home and to improve employees’ control over working time, shift scheduling, and when, where, and how to do their work. Among the total sample randomized, the intervention decreased the risk of burnout (*p* < 0.01). However, the intervention had beneficial effects on perceived stress (*p* < 0.05) and psychological distress (*p* < 0.05) only among the subgroup of workers who received intervention before the firm’s merger announcement, but not among workers who received intervention after the merger announcement. Changes in schedule control mediated the effect of intervention on psychological distress by 23% and on burnout by 19% among workers who received intervention before the firm’s merger announcement. However, this trial did not control the subgroup analyses for multiple testing, and the observed associations in the subgroup do not remain statistically significant if the estimates are adjusted for Bonferroni correction.

A non-randomized controlled trial [27] examined the effects of an organizational initiative (a natural experiment) on mental health. The experiment aimed to increase flexible working arrangements by allowing employees to be able to change their schedules or work location. The natural experiment did not directly lead to changes in emotional exhaustion or psychological distress at the follow-up [27]. However, the experiment increased schedule control and decreased negative work-home spillover, which were both associated with lower emotional exhaustion and psychological distress. There was no statistically significant difference in the associations between men and women [27].

A small cross-over study (N = 71) [16] found that the transition into open workspaces including flexible working arrangements reduces the risk of occupational stress; however, the beneficial effect was small. There was an interaction between flexible working arrangements and job autonomy. Flexible working arrangements reduced the risk of occupational stress in employees with high job autonomy but not in those with low job autonomy [16]. A small prospective cohort study (N = 91) [42] found no beneficial effect of flexibility idiosyncratic deals consisting of working time flexibility, work schedule flexibility and influence over working hours at baseline, on work engagement at follow-up. However, a prospective cohort study on flexible working arrangements consisting of (1) flexitime, (2) a compressed working week, (3) telecommuting, and (4) part-time work found lower work engagement and higher psychological distress at follow-up in workers who were provided flexible working arrangements at baseline [40]. However, the adverse effect on psychological distress was small.

## 4. Discussion

This systematic review aimed to assess the effects of employee-oriented flexible work on mental health. The findings indicate that flexible work increases employees’ control over working hours and has beneficial effects on depressive symptoms, burnout, fatigue, psychological distress, and emotional exhaustion. However, the beneficial effects are modest, and evidence is mostly based on observational studies. The review included only one randomized controlled trial and the rest were observational studies. Flexible work had no effect on work engagement; however, there were only two studies with work engagement as an outcome [17,42].

Flexible work can allow workers to fulfil both family and work responsibilities. Studies reported that women may perceive lower levels of control over daily working hours than men [34], and flexible work arrangements may have a larger beneficial effect on work-family conflict among women than men [43]. Furthermore, two other cohort studies found that low worktime control is associated with a higher risk of psychological distress [31] and disability pension due to mental health problems [30] in women only. However, a cohort study found no gender difference in the association between worktime control and depressive symptoms [34], and a cross-sectional study showed that low work flexibility is associated with a larger increase in emotional exhaustion in men than in women [44]. Employees with family responsibilities might benefit more from flexible work than employees without family responsibilities. Company policies toward flexibility may favour employees who justify taking advantage of flexible work, so lower perceived stress in employees with flexible work compared with those without flexible might be due to the fulfilling of family responsibilities.

Work-life imbalance may partly mediate the effect of worktime control on mental health [35]. Low worktime control can lead to work-to-family conflict [20], and work-to-family conflict can increase the risk of poor mental health [45]. Sleep disturbance may also play a role in the association between worktime control and mental health. This hypothesis has not been studied, but low control over taking breaks during work, handling private matters during work, and taking paid leave were associated with a higher rate of injuries at work or during leisure time, and accidents during commuting to and from work [46]. Sleep disturbance partially mediated the association between low worktime control and accidents [46]. However, the indirect effect of sleep disturbance on accidents was small (5%).

This is the first comprehensive review on the role of worktime flexibility, telework and flexible working arrangements in the prevention of mental health problems and their associated disability. We searched seven databases using comprehensive search terms and screened over 9393 abstracts. However, our search strings lacked a search term “worktime control”. We conducted additional searches for worktime control in relation to mental health problems in PubMed and Embase, and also searched the reference lists of reviews and original studies on worktime control. The studies included in the present review had some limitations. The studies used different definitions for flexible work. Some studies considered part-time work [40] and a compressed working week [40] as flexible work. However, people may work part-time because of a health problem, and in a compressed working week employees work nine hours or longer per day. Long working hours increase the risk of mental health problems [47], and a reduction in working time improves sleep duration and quality and reduces perceived stress [48]. This may be one of the reasons that a study found lower work engagement and higher psychological strain in employees who used flexible working arrangements that consisted of part-time work and a compressed working week [40]. Flexible work might also be arranged for workers with a health problem. The current evidence does not allow us to disentangle whether flexible work arrangements improve mental health or flexible work arrangements are a consequence of a mental health problem. Moreover, hybrid work might be more difficult to manage than solely remote or in-person work. This might contribute to adverse effects of flexible work arrangements.

Only a few studies examined the effects of flexible work on mental health problems, and the majority of the studies were observational ones, which are prone to selection bias and confounding. Further randomized controlled trials are needed to support the findings of observational studies on the associations between flexible work and mental health. Some of the observational studies did not control for some known confounding factors, thus the observed associations can be partly due to residual confounding. Moreover, some of the included studies recruited a small number of workers and did not have adequate statistical power to detect a modest effect of flexible work.

We did not include observational studies on self-scheduling or control over scheduling of shifts among shift workers which combined various shift workers (e.g., night, evening, or day workers) in a single analysis, did not control the health effect of scheduling flexibility for types of shifts, or did not conduct subgroup analysis by types of shifts. Shift work has adverse effects on mental health [49], and the health effects of control over scheduling of shifts can differ between different shift workers. The effect of scheduling flexibility on mental health should be estimated for different shift workers. Lastly, we did not use the GRADE tool [50] to rate the level of evidence. A small number of studies, mostly observational ones, examined the effects of various interventions/exposures on several mental health outcomes. The number of studies on each intervention/exposure-outcome pair was limited.

## 5. Conclusions

Worktime flexibility, working from home, and other flexible working arrangements may modestly improve self-rated mental health; however, the evidence is limited and based on observational studies with varying mental health outcomes. Intervention studies, particularly randomized and non-randomized controlled trials are needed to study the effect of flexible work, particularly working from home, on mental health. As the COVID-19 pandemic drastically increased working from home, this offers an opportunity to study the effects of working from home on mental health.

## Figures and Tables

**Table 1 healthcare-10-00883-t001:** Brief description of the characteristics of studies included in the review.

N of Studies	Study	Country	Follow-Up Time (Years)	Population	Sample	Age (Mean, Range, or %)	% of Female Sex	Exposure or Intervention	Outcome	Results
**Cluster randomized controlled trials**										
1	Moen 2016 [18]	USA	1	Employees and managers of a large firm’s IT division	865 (436 intervention group, 429 control group)	Birth year: 1946–1980	37.9	The intervention to facilitate working at home and to improve employees’ control over working time, shift scheduling, and when, where, and how to do their work.	Burnout, perceived stress, psychological distress	Among total sample, the intervention significantly decreased burnout.
**Non-randomized controlled trials**										
2	Moen 2011 [27]	USA	0.5	White-collar workers of corporate headquarters of Best Buy Co., Inc.	659 (325 intervention, 334 control)	32	48.4	The natural experiment moved employees from conventional practices to environments wherein they did not need permission to modify their work location or schedules.	Emotional exhaustion, psychological distress	Intervention did not directly change emotional exhaustion or psychological distress, but indirectly affected these outcomes by increasing schedule control and decreasing negative work-home spillover, which both improved well-being outcomes.
3	Nabe-Nielsen 2011 [28]	Denmark	1	Eldercare workers	35 intervention subgroup A, 187 controls	44 to 45	100	Worktime self-scheduling via a computer program (subgroup A)	Self-reported stress	The intervention increased employee involvement in planning of their working hours but did not decrease stress.
**Cross-over studies**										
4	Mache 2020 [16]	Germany	1	Full-time employees of a large technology company	71	39	53.5	Transition into open workspaces including flexible working arrangements	Occupational stress	Occupational stress decreased one year after flexible working arrangements.
5	Vesala 2015 [17]	Finland	0.1 to 0.25	A sample of knowledge workers	39	44	50	One week telework in the rural archipelago environment	Stress, emotional exhaustion, and work engagement	Stress reduced during the telework period and did not reach the original level after experiment. Emotional exhaustion reduced during telework but returned to the original level after experiment. Telework had no effect on work engagement.
**Prospective cohort studies**										
6	Albrecht 2020a [29]	Finland	7	Full-time employees of public sector	22599	39% were ≥50 years	75	Worktime control	Sickness absence due to depression or anxiety	Control over daily hours and control over time off were not associated with sickness absence due to depression or anxiety.
6	Vahtera 2010 [30]	Finland	4.4	Employees of public sector	30700	44.8	77.5	Worktime control	Disability pension due to mental disorders	Self-assessed, but not co-worker assessed worktime control was associated with lower risk of disability pension among women.
6	Ala-Mursula 2004 [31]	Finland	3	Permanent full-time employees of public sector	4218	Men 46, women 45	78.6	Worktime control	Psychological distress	Low worktime control was related to high psychological distress in women.
7	Aronsson 2019 [32]	Sweden	2	General working population who worked at least 30% full-time	4408	51	58	Worktime control	Burnout	Population attributable fraction of burnout for low worktime control was 5% for human service occupations and 7% for other occupations.
8	Lee 2018 [33]	USA	1.5	Employees from 26 different technology offices	507	Not reported	45	Control over working hours/schedule	Burnout	A positive correlation between low control over working hours and burnout.
9	Albrecht 2017 [34]	Sweden	6	General working population	2722	47	58.6	Worktime control	Depressive symptoms	Low control over daily hours and low control over time off were associated with higher subsequent depressive symptoms.
9	Albrecht 2020b [35]	Sweden	6	General working population	26804	49	55.4	Worktime control	Depressive symptoms	Work-life imbalance partially mediated the relationship between worktime control and depressive symptoms.
10	Windeler 2017 [36]	USA	0.3	Employees of the IT business unit of a financial services firm	51	43	39	Teleworking for 1–2 days per week	Work exhaustion	Work exhaustion increased as interpersonal interaction increased. Part-time telework reduced the effect of interpersonal interaction on work exhaustion.
11	Henke 2016 [37]	USA	2	Active prudential financial employees	3703	88% were <55 years	62	Prime time telecommuters, off-hour telecommuters	Depression, stress	A U-shaped or J-shaped association between the number of hours worked from home per month and depression. No association with stress.
12	Kubo 2016 [38]	Japan	1	Employees of a manufacturing industry and a research institute	37	41.9	23	Worktime control	Subjectively assessed accumulated fatigue and objectively measured fatigue	Increase in worktime control during one-year follow-up was not associated with accumulated fatigue but had positive effect on objectively measured fatigue.
13	Shepherd-Banigan 2016 [39]	USA	2	Working women who returned to work within six months after childbirth	570	29.6	100	Schedule flexibility, working from home	Depressive symptoms	Working from home reduced depressive symptoms, but schedule flexibility and number of hours worked from home were not associated with changes in depressive symptoms.
14	Timms 2015 [40]	Australia	1	Employees representing education, banking, and public/community services	823	43	72	Flexible working arrangements	Psychological strain work engagement	Use of flexible work arrangement was associated with lower work engagement and higher psychological strain.
15	Takahashi 2012 [41]	Japan	1.3	Daytime managers, professionals, and clerical, sales or transportation workers	2382	40.6	34.8	Worktime control	Fatigue, depressive symptoms	High worktime control was related to lower levels of fatigue and depressive symptoms.
16	Hornung 2011 [42]	Germany	1	Medical doctors	91	39.4	47	Flexibility idiosyncratic deals: (1) working time flexibility, (2) work schedule flexibility, and (3) influence over working hours.	Work-family conflict, work-related well-being (work engagement).	Idiosyncratic deals related were not associated with work-family conflict or with work engagement.

**Figure 1 healthcare-10-00883-f001:**
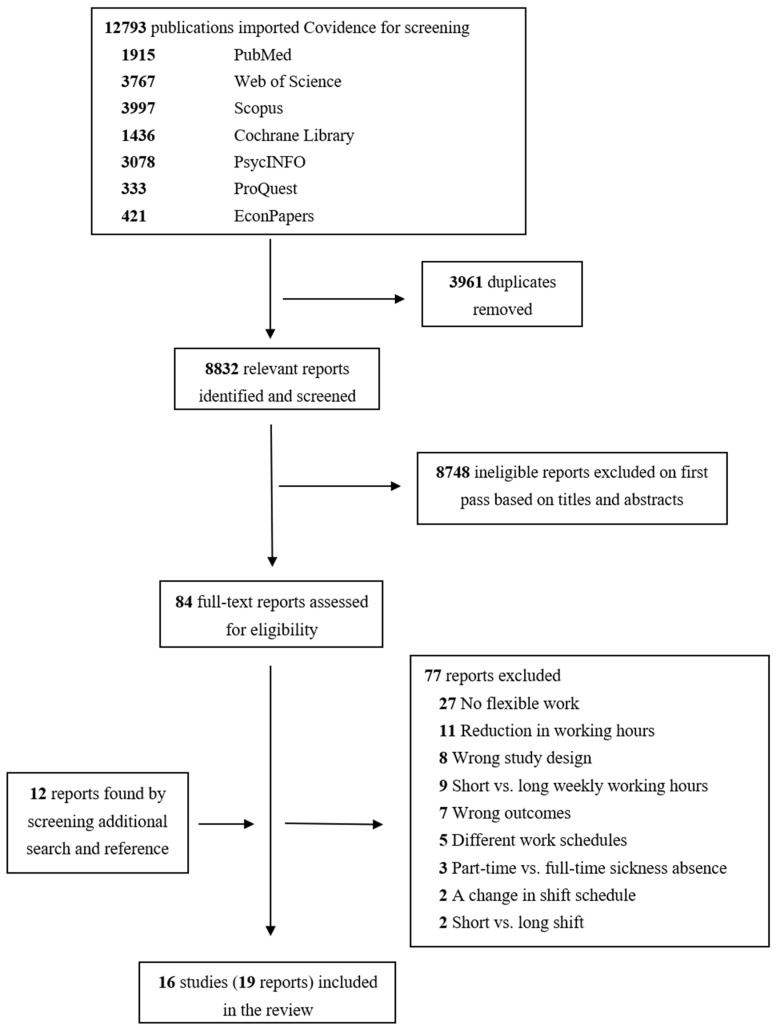
PRISMA flow diagram of the studies selection.

## Data Availability

Not applicable.

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
