# Peer review of "The Effect of Employee-Oriented Flexible Work on Mental Health: A Systematic Review"

_healthcare, 2022, doi:10.3390/healthcare10050883_

Round 1

Reviewer 1 Report

I feel that this article makes a solid contribution to the literature and is publication worthy.

I would point out the following for consideration:

(1) It has been said that "hybrid" work is more difficult for management to manage as opposed to 100% in-person or 100% remote.  So, I am wondering if managers might be implicitly conveying their discomfort with their employees' hybrid work arrangements and somehow contribute to perceived stress by direct reports, thereby neutralizing the benefits of "control" that these staff have over their work.

(2) In addition, not all employees have the same personal situations.  I would think that those with significant non-work and family responsibilities might benefit Psychologically and pragmatically more from the flexibility than those with less personal responsibilities.  So, there might be an equity issue when company policies toward flexibility implicitly favor those who have greater "justification" for taking advantage of that flexibility.  So, perceived stress might emanate from that phenomenon in a differential way as an unexpected artifact.

Author Response

Reviewer 1

Comments and Suggestions for Authors

I feel that this article makes a solid contribution to the literature and is publication worthy.

I would point out the following for consideration:

Response. Thank you for your comments!

(1) It has been said that "hybrid" work is more difficult for management to manage as opposed to 100% in-person or 100% remote.  So, I am wondering if managers might be implicitly conveying their discomfort with their employees' hybrid work arrangements and somehow contribute to perceived stress by direct reports, thereby neutralizing the benefits of "control" that these staff have over their work.

Response. We have discussed this issue on page 21, the last paragraph, as follows:

“Moreover, hybrid work might be more difficult to manage than solely remote or in-person work. This might contribute to adverse effects of flexible work arrangements.”

(2) In addition, not all employees have the same personal situations.  I would think that those with significant non-work and family responsibilities might benefit Psychologically and pragmatically more from the flexibility than those with less personal responsibilities.  So, there might be an equity issue when company policies toward flexibility implicitly favor those who have greater "justification" for taking advantage of that flexibility.  So, perceived stress might emanate from that phenomenon in a differential way as an unexpected artifact.

Response. We have discussed this issue on page 20, the second paragraph, as follows:

“Employees with family responsibilities might benefit more from flexible work than employees without family responsibilities. Company policies toward flexibility may favour employees who justify taking advantage of flexible work, so lower perceived stress in employees with flexible work compared with those without flexible might be due to fulfilling family responsibilities.”

Reviewer 2 Report

The authors conducted a systematic review of the effect of flexible work on mental health.  This topic is relevant following changes in work structure after the pandemic.  I appreciate the effort of the authors and hope my feedback is beneficial.

-In the introduction, more clarification is needed as to what this systematic review adds to the current literature base.  The authors cited three systematic reviews.  It is important to better explain the unique contribution of this systematic review, or how it differs from existing research.  

-In the introduction, it would be helpful to clarify what the authors mean by "flexible work in the prevention of mental health problems."  This does not seem to be what they actually reviewed (which was research on flexible work schedules and different mental health outcomes).  

-In Methods, it would be helpful to describe the PRISMA statement that was used to develop the review protocol  

-More clarity is needed about the search terms used

-More description of inclusion and exclusion criteria is needed.

-The authors noted that "randomized and non-randomized clinical trials, cross-over studies, prospective cohort studies, and case-control studies were eligible.  Cross-sectional studies were excluded due to their descriptive nature."  Yet when discussing results later, the authors report that the conclusions made by the research reviewed were largely based on observational studies.  Please clarify.

-Please include the percentage of interrater agreement between the reviewers' screenings

-It would be helpful to briefly describe the Cochrane risk of bias tool for readers who may be unfamiliar

-Under data synthesis, what specific statistical methods were used to "synthesize the results qualitatively"?

-Under Results, please further describe what criteria was used to screen titles and abstracts.

-When discussing studies included in the review, it would be helpful to more specifically explain what "items" the studies used for assessment.  Similarly, it will be helpful to explain what is meant by "mental health outcomes" as this can mean different things (depression? functioning? quality of life? symptoms of other diagnoses?)

-In general, more critical appraisal and synthesis of the research reviewed would be helpful.  As written, the paper vaguely summarizes studies and is lacking conclusions drawn from these studies as a whole

-In the discussion, the authors note that this is the "first comprehensive review on the role of worktime flexibility, telework and flexible working arrangements in the prevention of mental health problems and their associated disability".  However, it does not appear that any conclusions were made regarding this aim.  This will be a helpful section for the authors to make summary points and recommendations for future research/directions.  What does this review say about work structures and mental health?

-It seemed that the results of the research reviewed were quite variable, yet the authors did not comment to this point.  I'm curious about their thoughts.

-In the conclusion section, the authors remarked that "Intervention studies are needed to study the effect of flexible work, particularly working from home, on mental health."  It would be helpful to explain more about what is meant by intervention studies.  Also, it would be helpful to expand upon this conclusion in the discussion section.

Author Response

Reviewer 2

Comments and Suggestions for Authors

The authors conducted a systematic review of the effect of flexible work on mental health.  This topic is relevant following changes in work structure after the pandemic.  I appreciate the effort of the authors and hope my feedback is beneficial.

Response. Thank you for your comments!

-In the introduction, more clarification is needed as to what this systematic review adds to the current literature base.  The authors cited three systematic reviews.  It is important to better explain the unique contribution of this systematic review, or how it differs from existing research.  

Response. In the earlier reviews, the association between employee-oriented flexible work and mental health was mainly based on cross-sectional studies. Furthermore, the previous reviews did not examine the association between working from home and mental health. We have explained this on page 6, the first paragraph, as follows:

“In the earlier reviews, the association between employee-oriented flexible work and mental health was mainly based on cross-sectional studies. Furthermore, the previous reviews did not examine the association between working from home and mental health.”

-In the introduction, it would be helpful to clarify what the authors mean by "flexible work in the prevention of mental health problems."  This does not seem to be what they actually reviewed (which was research on flexible work schedules and different mental health outcomes).  

Response. We have clarified flexible work on page 6, the last paragraph, as follows:

“In the present study, we aimed to systematically review available literature on the effects of employee-oriented worktime control, working from home, flexible work (when, where, or how to do the work), and employee-oriented flexible workplace arrangements on mental health, mental disorders, and work disability due to mental disorders.”

-In Methods, it would be helpful to describe the PRISMA statement that was used to develop the review protocol  

-More clarity is needed about the search terms used

-More description of inclusion and exclusion criteria is needed.

Response. We have explained the PRISMA statement on page 7, the 2nd paragraph, as follows:

“The checklist consists of 27 recommendations which help the reviewers to report transparently the results of their systematic reviews.”

We have clarified search terms used in the different databases on page 7, the 2nd paragraph.

“Supplemental Table 1 shows the search terms and search strings used in the different databases.”

We have also clarified the inclusion and exclusion criteria on page 7, the last paragraph and page 8, the 1st paragraph.

-The authors noted that "randomized and non-randomized clinical trials, cross-over studies, prospective cohort studies, and case-control studies were eligible.  Cross-sectional studies were excluded due to their descriptive nature."  Yet when discussing results later, the authors report that the conclusions made by the research reviewed were largely based on observational studies.  Please clarify.

Response. Observational studies include case series, cross-sectional studies, case-control studies, cohort studies, cross-over studies (without randomization), uncontrolled before-and-after studies, and non-randomized clinical trials. The current review included only one randomized controlled trial. We have clarified this on page 20, first paragraph, as follows:

“The review included only one randomized controlled trial and the rest were observational studies.”

-Please include the percentage of interrater agreement between the reviewers' screenings

-It would be helpful to briefly describe the Cochrane risk of bias tool for readers who may be unfamiliar

Response. We have added the overall percentage of interrater agreement between the reviewers on page 9, the 2nd paragraph, and described the Cochrane risk of bias tool on page 8, the 2nd paragraph, as follows:

“The overall percentage of interrater agreement between the reviewers was more than 97%.”

“The Cochrane risk of bias tool (ROB-2) evaluates bias arising from the randomization process, bias arising from the timing of randomization, bias due to deviations from intended interventions, bias due to missing outcome data, bias in measurement of the outcome, and bias in selection of the reported result.”

-Under data synthesis, what specific statistical methods were used to "synthesize the results qualitatively"?

Response. There are two types of systematic reviews: systematic review with quantitative synthesis and systematic review with qualitative synthesis. The first one includes a meta-analysis, and the latter includes only descriptive (qualitative) synthesis. A meta-analysis was not possible in the current review, and we reported only descriptive results synthesized qualitatively.

-Under Results, please further describe what criteria was used to screen titles and abstracts.

Response. The inclusion/exclusion criteria were used for the literature searches and screening titles/abstracts. The titles/abstracts were screened for study design, population, exposures/interventions, and outcomes. We have clarified this on page 9, the 2nd paragraph, as follows:

“A total of 8748 reports were ineligible based on screening titles and abstracts for study design, population, exposures/interventions, and outcomes.”

-When discussing studies included in the review, it would be helpful to more specifically explain what "items" the studies used for assessment.  Similarly, it will be helpful to explain what is meant by "mental health outcomes" as this can mean different things (depression? functioning? quality of life? symptoms of other diagnoses?)

Response. We have listed mental health outcomes on page 9, last paragraph, as follows:

“Mental health outcomes included depressive symptoms, perceived stress, psychological distress, emotional exhaustion, burnout, accumulated fatigue, disability pension due to mental disorders, sickness absence due to depression or anxiety, and work-related well-being (work engagement).”

-In general, more critical appraisal and synthesis of the research reviewed would be helpful.  As written, the paper vaguely summarizes studies and is lacking conclusions drawn from these studies as a whole

Response. We did not use the GRADE tool to rate the level of evidence. A small number of studies, mostly observational studies examined the effects of various interventions/exposures on several mental health outcomes. The number of studies on each intervention/exposure - outcome pair was limited. We have discussed this on page 22, the 2nd paragraph, as follows:

“Lastly, we did not use the GRADE tool [51] to rate the level of evidence. A small number of studies, mostly observational studies examined the effects of various interventions/exposures on several mental health outcomes. The number of studies on each intervention/exposure - outcome pair was limited.”

-In the discussion, the authors note that this is the "first comprehensive review on the role of worktime flexibility, telework and flexible working arrangements in the prevention of mental health problems and their associated disability".  However, it does not appear that any conclusions were made regarding this aim.  This will be a helpful section for the authors to make summary points and recommendations for future research/directions.  What does this review say about work structures and mental health?

Response. We did not study the association between work structures and mental health, that is, work structures were not part of our inclusion and exclusion criteria, and we did not search the databases for these types of the studies. However, we have slightly modified our conclusions, as follows:

“Worktime flexibility, working from home, and other flexible working arrangements may modestly improve self-rated mental health, however, the evidence is limited and based on observational studies with varying mental health outcomes. Intervention studies, particularly randomized and non-randomized controlled trials are needed to study the effect of flexible work, particularly working from home, on mental health. As the COVID-19 pandemic drastically increased working from home, this offers an opportunity to study the effects of working from home on mental health.”

-It seemed that the results of the research reviewed were quite variable, yet the authors did not comment to this point.  I'm curious about their thoughts.

Response. The included studies differed in study design, intervention/exposure (worktime control, working from home/teleworking, or a combination of flexible working arrangements) and outcome (different indicators of symptoms of mental ill-health or positive mental health outcomes). Thus, we were not able to conduct a meta-analysis, and synthesized the results qualitatively. We have added the following to conclusions:

“Worktime flexibility, working from home, and other flexible working arrangements may modestly improve self-rated mental health, however, the evidence is limited and based on observational studies with varying mental health outcomes.”

-In the conclusion section, the authors remarked that "Intervention studies are needed to study the effect of flexible work, particularly working from home, on mental health."  It would be helpful to explain more about what is meant by intervention studies.  Also, it would be helpful to expand upon this conclusion in the discussion section.

Response. Intervention studies include randomized controlled trials, non-randomized controlled trials, before–after (pre–post) studies, crossover studies, and intervention studies without concurrent controls (with historical controls). We have clarified this on page 22, last paragraph, and page 23, the first paragraph, as follows:

“Intervention studies, particularly randomized and non-randomized controlled trials are needed to study the effect of flexible work, particularly working from home, on mental health.”

We also discussed this issue on page 22, first paragraph, as follows:

“Further randomized controlled trials are needed to support the findings of observational studies on the associations between flexible work and mental health.”

Reviewer 3 Report

This work is well written, used PRISMA guidelines, but I think it also has important things to consider: 

This study found lower work engagement and higher psychological strain in employees who used flexible working arrangements that consisted of part-time work and compressed working weeks. There was diversity in different definitions for flexible work. My consideration is to find at least other 1000 articles for the review and try to standardize and make at least a partial conclusion, even in a review need.

I also find the 3.3. Flexible working arrangements section weak. You should explain it better and make it clear. 

Author Response

Reviewer 3

Comments and Suggestions for Authors

This work is well written, used PRISMA guidelines, but I think it also has important things to consider: 

Response. Thank you for your comments!

This study found lower work engagement and higher psychological strain in employees who used flexible working arrangements that consisted of part-time work and compressed working weeks. There was diversity in different definitions for flexible work. My consideration is to find at least other 1000 articles for the review and try to standardize and make at least a partial conclusion, even in a review need.

Response. We have searched multiple databases using sensitive search terms and screened 9393 (8832 + 561) publications. It is unlikely that new searches identify more studies. One or two new studies do not change the conclusions of the review. After 2-3 years, sufficient new publications may be available and allows updating the review. The included studies examined different mental health outcomes, so we did not conduct any quantitative analysis.

I also find the 3.3. Flexible working arrangements section weak. You should explain it better and make it clear. 

Response. We have revised this section.

Round 2

Reviewer 2 Report

Thank you for your attention to these comments.  The clarity in purpose and critique is improved.

Reviewer 3 Report

Thank you to all authors for taking your time to meticulously correct all the typos that were mentioned.